# Dynamic Graph Neural Networks Under Spatio-Temporal Distribution Shift

**Zeyang Zhang[1]**\*, **Xin Wang[1]**†, **Ziwei Zhang[1]**, **Haoyang Li[1]**, **Zhou Qin[2]**, **Wenwu Zhu[1]**†

[1]Tsinghua University, [2]Alibaba Group

zy-zhang20@mails.tsinghua.edu.cn, {xin_wang, zwzhang}@tsinghua.edu.cn,
lihy18@mails.tsinghua.edu.cn, qinzhou.qinzhou@alibaba-inc.com,
wwzhu@tsinghua.edu.cn

## Abstract

Dynamic graph neural networks (DyGNNs) have demonstrated powerful predictive abilities by exploiting graph structural and temporal dynamics. However, the existing DyGNNs fail to handle distribution shifts, which naturally exist in dynamic graphs, mainly because the patterns exploited by DyGNNs may be variant with respect to labels under distribution shifts. In this paper, we propose to handle spatio-temporal distribution shifts in dynamic graphs by discovering and utilizing *invariant patterns*, i.e., structures and features whose predictive abilities are stable across distribution shifts, which faces two key challenges: 1) How to discover the complex variant and invariant spatio-temporal patterns in dynamic graphs, which involve both time-varying graph structures and node features. 2) How to handle spatio-temporal distribution shifts with the discovered variant and invariant patterns. To tackle these challenges, we propose the Disentangled Intervention-based Dynamic graph Attention networks (**DIDA**). Our proposed method can effectively handle spatio-temporal distribution shifts in dynamic graphs by discovering and fully utilizing invariant spatio-temporal patterns. Specifically, we first propose a disentangled spatio-temporal attention network to capture the variant and invariant patterns. Then, we design a spatio-temporal intervention mechanism to create multiple interventional distributions by sampling and reassembling variant patterns across neighborhoods and time stamps to eliminate the spurious impacts of variant patterns. Lastly, we propose an invariance regularization term to minimize the variance of predictions in intervened distributions so that our model can make predictions based on invariant patterns with stable predictive abilities and therefore handle distribution shifts. Experiments on three real-world datasets and one synthetic dataset demonstrate the superiority of our method over state-of-the-art baselines under distribution shifts. Our work is the first study of spatio-temporal distribution shifts in dynamic graphs, to the best of our knowledge.

## 1 Introduction

Dynamic graphs widely exist in real-world applications, including financial networks [1, 2], social networks [3, 4], traffic networks [5, 6], etc. Distinct from static graphs, dynamic graphs can represent temporal structure and feature patterns, which are more complex yet common in reality. Dynamic graph neural networks (DyGNNs) have been proposed to tackle highly complex structural and temporal information over dynamic graphs, and have achieved remarkable progress in many predictive tasks [7, 8].

---

\*This work was done during author's internship at Alibaba Group
†Corresponding authors

36th Conference on Neural Information Processing Systems (NeurIPS 2022).

Nevertheless, the existing DyGNNs fail to handle spatio-temporal distribution shifts, which naturally exist in dynamic graphs for various reasons such as survivorship bias [9], selection bias [10, 11], trending [12], etc. For example, in financial networks, external factors like period or market would affect the correlations between the payment flows and transaction illegitimacy [13]. Trends or communities also affect interaction patterns in coauthor networks [14] and recommendation networks [15]. If DyGNNs highly rely on spatio-temporal patterns which are variant under distribution shifts, they will inevitably fail to generalize well to the unseen test distributions.

To address this issue, in this paper, we study the problem of handling spatio-temporal distribution shifts in dynamic graphs through discovering and utilizing *invariant patterns*, i.e., structures and features whose predictive abilities are stable across distribution shifts, which remain unexplored in the literature. However, this problem is highly non-trivial with the following challenges:

- How to discover the complex variant and invariant spatio-temporal patterns in dynamic graphs, which include both graph structures and node features varying through time?

- How to handle spatio-temporal distribution shifts in a principled manner with discovered variant and invariant patterns?

To tackle these challenges, we propose a novel DyGNN named Disentangled Intervention-based Dynamic Graph Attention Networks (**DIDA**[3]). Our proposed method handles distribution shifts well by discovering and utilizing invariant spatio-temporal patterns with stable predictive abilities. Specifically, we first propose a disentangled spatio-temporal attention network to capture the variant and invariant patterns in dynamic graphs, which enables each node to attend to all its historic neighbors through a disentangled attention message-passing mechanism. Then, inspired by causal inference literatures [16, 17], we propose a spatio-temporal intervention mechanism to create multiple intervened distributions by sampling and reassembling variant patterns across neighborhoods and time, such that spurious impacts of variant patterns can be eliminated. To tackle the challenges that i) variant patterns are highly entangled across nodes and ii) directly generating and mixing up subsets of structures and features to do intervention is computationally expensive, we approximate the intervention process with summarized patterns obtained by the disentangled spatio-temporal attention network instead of original structures and features. Lastly, we propose an invariance regularization term to minimize prediction variance in multiple intervened distributions. In this way, our model can capture and utilize invariant patterns with stable predictive abilities to make predictions under distribution shifts. Extensive experiments on one synthetic dataset and three real-world datasets demonstrate the superiority of our proposed method over state-of-the-art baselines under distribution shifts. The contributions of our work are summarized as follows:

- We propose Disentangled Intervention-based Dynamic Graph Attention Networks (**DIDA**), which can handle spatio-temporal distribution shifts in dynamic graphs. This is the first study of spatio-temporal distribution shifts in dynamic graphs, to the best of our knowledge.

- We propose a disentangled spatio-temporal attention network to capture variant and invariant graph patterns. We further design a spatio-temporal intervention mechanism to create multiple intervened distributions and an invariance regularization term based on causal inference theory to enable the model to focus on invariant patterns under distribution shifts.

- Experiments on three real-world datasets and one synthetic dataset demonstrate the superiority of our method over state-of-the-art baselines.

## 2   Problem Formulation

In this section, we formulate the problem of spatio-temporal distribution shift in dynamic graphs.

**Dynamic Graph.** Consider a graph $\mathcal{G}$ with the node set $\mathcal{V}$ and the edge set $\mathcal{E}$. A dynamic graph can be defined as $\mathcal{G} = (\{\mathcal{G}^t\}_{t=1}^T)$, where $T$ is the number of time stamps, $\mathcal{G}^t = (\mathcal{V}^t, \mathcal{E}^t)$ is the graph slice at time stamp $t$, $\mathcal{V} = \bigcup_{t=1}^T \mathcal{V}^t$, $\mathcal{E} = \bigcup_{t=1}^T \mathcal{E}^t$. For simplicity, a graph slice is also denoted as $\mathcal{G}^t = (\mathbf{X}^t, \mathbf{A}^t)$, which includes node features and adjacency matrix at time $t$. We use $\mathbf{G}^t$ to denote a random variable of $\mathcal{G}^t$.

---

[3]Our codes are publicly available at https://github.com/wondergo2017/DIDA

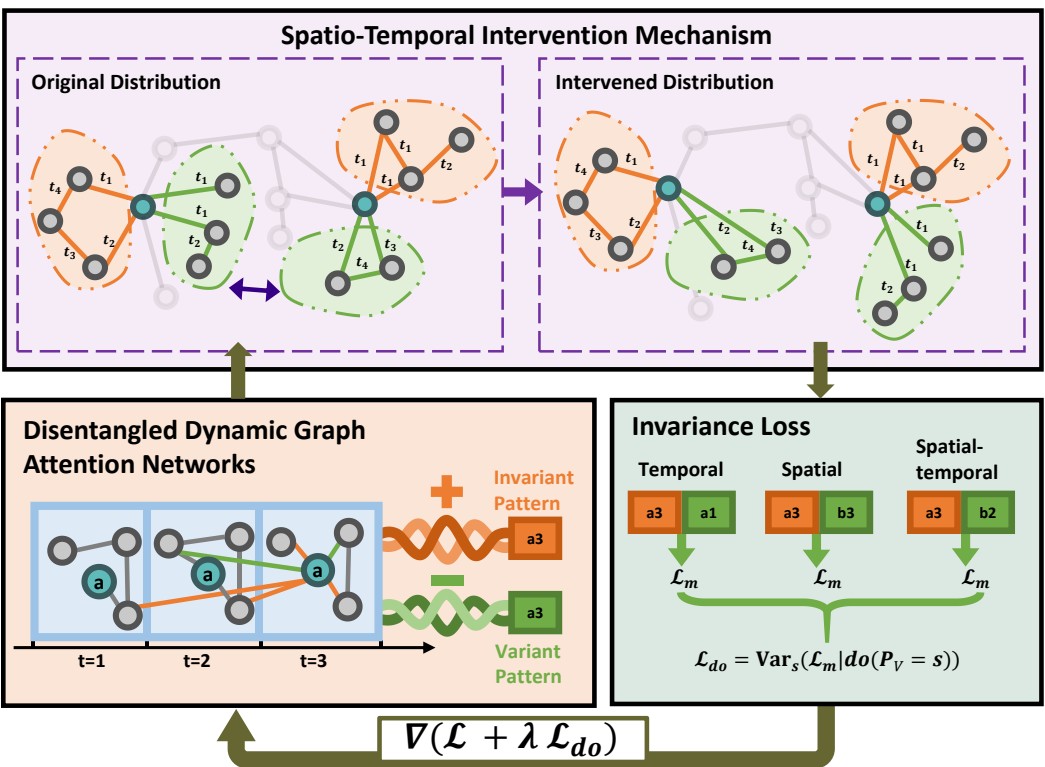

Figure 1: The framework of our proposed method **DIDA**. (Bottom left) For a given dynamic graph with multiple timestamps, the disentangled dynamic graph attention networks first obtain summarizations of high-order invariant and variant patterns by disentangled spatio-temporal message passing. (Top) Then the spatio-temporal intervention mechanism creates multiple intervened distributions by sampling and reassembling variant patterns across space and time for each node. (Bottom right) Last, invariance loss is calculated by using samples from intervened distributions to optimize the model so that it can focus on invariant patterns to make predictions.

**Prediction tasks.** For dynamic graphs, the prediction task can be summarized as using past graphs to make predictions, i.e. $p(\mathbf{Y}^t|\mathbf{G}^1, \mathbf{G}^2, \dots, \mathbf{G}^t){=}p(\mathbf{Y}^t|\mathbf{G}^{1:t})$ , where label $\mathbf{Y}^t$ can be node properties or occurrence of links between nodes at time $t+1$. In this paper, we mainly focus on node-level tasks, which are commonly adopted in dynamic graph literatures [7, 8]. Following [18, 19], we factorize the distribution of graph trajectory into ego-graph trajectories, i.e. $p(\mathbf{Y}^t \mid \mathbf{G}^{1:t}) = \prod_v p(\mathbf{y}^t \mid \mathbf{G}_v^{1:t})$. An ego-graph induced from node $v$ at time $t$ is defined as $\mathcal{G}_v^t = (\mathbf{X}_v^t, \mathbf{A}_v^t)$ where $\mathbf{A}_v^t$ is the adjacency matrix including all edges in node $v$'s $L$-hop neighbors at time $t$, i.e. $\mathcal{N}_v^t$, and $\mathbf{X}_v^t$ includes the features of nodes in $\mathcal{N}_v^t$. The optimization objective is to learn an optimal predictor with empirical risk minimization

$$\min_\theta \mathbb{E}_{(y^t, \mathcal{G}_v^{1:t}) \sim p_{tr}(\mathbf{y}^t, \mathbf{G}_v^{1:t})} \mathcal{L}(f_\theta(\mathcal{G}_v^{1:t}), y^t) \tag{1}$$

where $f_\theta$ is a learnable dynamic graph neural networks, We use $\mathbf{G}_v^{1:t}, \mathbf{y}^t$ to denote the random variable of the ego-graph trajectory and its label, and $\mathcal{G}_v^{1:t}, y^t$ refer to the respective instances.

**Spatio-temporal distribution shift.** However, the optimal predictor trained with the training distribution may not generalize well to the test distribution when there exists a distribution shift problem. In the literature of dynamic graph, researchers are devoted to capture laws of network dynamics which are stable in systems [20, 21, 22, 23, 24]. Following them, we assume the conditional distribution is the same $p_{tr}(\mathbf{Y}^t|\mathbf{G}^{1:t}) = p_{te}(\mathbf{Y}^t|\mathbf{G}^{1:t})$, and only consider the covariate shift problem where $p_{tr}(\mathbf{G}^{1:t}) \neq p_{te}(\mathbf{G}^{1:t})$. Besides temporal distribution shift which naturally exists in time-varying data [25, 12, 26, 27, 28] and structural distribution shift in non-euclidean data [29, 18, 30], there exists a much more complex spatio-temporal distribution shift in dynamic graphs. For example, the distribution of ego-graph trajectories may vary across periods or communities.

# 3 Method

In this section, we propose Disentangled Intervention-based Dynamic Graph Attention Networks (**DIDA**) to handle spatio-temporal distribution shift in dynamic graphs. First, we propose a disentangled dynamic graph attention network to extract invariant and variant spatio-temporal patterns. Then we propose a spatio-temporal intervention mechanism to create multiple intervened data distributions. Finally, we optimize the model with invariance loss to make predictions relying on invariant patterns.

## 3.1 Handling Spatio-Temporal Distribution Shift

**Spatio-Temporal Pattern.** In recent decades of development of dynamic graphs, some scholars endeavor to conclude insightful patterns of network dynamics to reflect how real-world networks evolve through time [31, 32, 33, 34]. For example, the laws of triadic closure describe that two nodes with common neighbors (patterns) tend to have future interactions in social networks [35, 36, 23]. Besides structural information, node attributes are also an important part of the patterns, e.g., social interactions can be also affected by gender and age [37]. Instead of manually concluding patterns, we aim at learning the patterns using DyGNNs so that the more complex spatio-temporal patterns with mixed features and structures can be mined in dynamic graphs. Therefore, we define the spatio-temporal pattern used for node-level prediction as a subset of ego-graph trajectory

$$P^t(v) = m_v^t(\mathcal{G}_v^{1:t}) \tag{2}$$

where $m_v^t(\cdot)$ selects structures and attributes from the ego-graph trajectory. In [23], the pattern can be explained as an open triad with similar neighborhood, and the model tend to make link predictions to close the triad with $\hat{y}_{u,v}^t = f_\theta(P^t(u), P^t(v))$ based on the laws of triadic closure [38]. DyGNNs aim at exploiting predictive spatio-temporal patterns to boost prediction ability. However, the predictive power of some patterns may vary across periods or communities due to spatio-temporal distribution shift. Inspired by the causal theory [16, 17], we make the following assumption

**Assumption 1** *For a given task, there exists a predictor $f(\cdot)$, for samples $(\mathcal{G}_v^{1:t}, y^t)$ from any distribution, there exists an invariant pattern $P_I^t(v)$ and a variant pattern $P_V^t(v)$ such that $y^t = f(P_I^t(v)) + \epsilon$ and $P_I^t(v) = \mathcal{G}_v^{1:t} \backslash P_V^t(v)$, i.e., $\mathbf{y}^t \perp \mathbf{P}_V^t(v) \mid \mathbf{P}_I^t(v)$.*

Assumption 1 shows that invariant patterns $\mathbf{P}_I^t(v)$ are sufficiently predictive for label $y^t$ and can be exploited across periods and communities without adjusting the predictor, while the influence of variant patterns $\mathbf{P}_V^t(v)$ on $\mathbf{y}^t$ is shielded by the invariant patterns.

**Training Objective.** Our main idea is that to obtain better generalization ability, the model should rely on invariant patterns instead of variant patterns, as the former is sufficient for prediction while the predictivity of the latter could be variant under distribution shift. Along this, our objective can be transformed to

$$\min_{\theta_1, \theta_2} \mathbb{E}_{(y^t, \mathcal{G}_v^{1:t}) \sim p_{tr}(\mathbf{y}^t, \mathbf{G}_v^{1:t})} \mathcal{L}(f_{\theta_1}(\tilde{P}_I^t(v)), y^t)$$
$$s.t \quad \phi_{\theta_2}(\mathcal{G}_v^{1:t}) = \tilde{P}_I^t(v), \mathbf{y}^t \perp \tilde{\mathbf{P}}_V^t(v) \mid \tilde{\mathbf{P}}_I^t(v). \tag{3}$$

where $f_{\theta_1}(\cdot)$ make predictions based on the invariant patterns, $\phi_{\theta_2}(\cdot)$ aims at finding the invariant patterns. Backed by causal theory[16, 17], it can be transformed into

$$\min_{\theta_1, \theta_2} \mathbb{E}_{(y^t, \mathcal{G}_v^{1:t}) \sim p_{tr}(\mathbf{y}^t, \mathbf{G}_v^{1:t})} \mathcal{L}(f_{\theta_1}(\phi_{\theta_2}(\mathcal{G}_v^{1:t})), y^t) +$$
$$\lambda \text{Var}_{s \in \mathcal{S}} (\mathbb{E}_{(y^t, \mathcal{G}_v^{1:t}) \sim p_{tr}(\mathbf{y}^t, \mathbf{G}_v^{1:t} \mid \text{do}(\mathbf{P}_V^t = s))} \mathcal{L}(f_{\theta_1}(\phi_{\theta_2}(\mathcal{G}_v^{1:t})), y^t)) \tag{4}$$

where 'do' denotes do-calculas to intervene the original distribution [39, 17], $\mathcal{S}$ denotes the intervention set and $\lambda$ is a balancing hyperparameter. The idea can be informally described that as in Eq. (3), variant patterns $\mathbf{P}_V^t$ have no influence on the label $\mathbf{y}^t$ given the invariant patterns $\mathbf{P}_I^t$, then the prediction would not be varied if we intervene the variant patterns and keep invariant patterns untouched. More details about the connections between objective Eq.(3) and Eq.(4) can be found in Appendix.

**Remark 1** *Minimizing the variance term in Eq. (4) help the model to satisfy the constraint of $\mathbf{y}^t \perp \tilde{\mathbf{P}}_V^t(v) \mid \tilde{\mathbf{P}}_I^t(v)$ in Eq. (3), i.e., $p(\mathbf{y}^t \mid \tilde{\mathbf{P}}_I^t(v), \tilde{\mathbf{P}}_V^t(v)) = p(\mathbf{y}^t \mid \tilde{\mathbf{P}}_I^t(v))$*

## 3.2 Disentangled Dynamic Graph Attention Networks

**Dynamic Neighborhood**. To simultaneously consider the spatio-temporal information, we define the dynamic neighborhood as $\mathcal{N}^t(u) = \{v : (u, v) \in \mathcal{E}^t\}$, which includes all nodes that have interactions with node $u$ at time $t$.

**Disentangled Spatio-temporal Graph Attention Layer.** To capture spatio-temporal pattern for each node, we propose a spatio-temporal graph attention to enable each node to attend to its dynamic neighborhood simultaneously. For a node $u$ at time stamp $t$ and its neighbors $v \in \mathcal{N}^{t'}(u), \forall t' \leq t$, we calculate the Query-Key-Value vectors as:

$$\mathbf{q}_u^t = \mathbf{W}_q(\mathbf{h}_u^t || \text{TE}(t)), \mathbf{k}_v^{t'} = \mathbf{W}_k(\mathbf{h}_v^{t'} || \text{TE}(t')), \mathbf{v}_v^{t'} = \mathbf{W}_v(\mathbf{h}_v^{t'} || \text{TE}(t')) \tag{5}$$

where $\mathbf{h}_u^t$ denotes the representation of node $u$ at the time stamp $t$, $\mathbf{q}$, $\mathbf{k}$, $\mathbf{v}$ represents the query, key and value vector, respectively, and we omit the bias term for brevity. $\text{TE}(t)$ denotes temporal encoding techniques to obtain embeddings of time $t$ so that the time of link occurrence can be considered inherently [40, 41]. Then, we can calculate the attention scores among nodes in the dynamic neighborhood to obtain the structural masks

$$\mathbf{m}_I = \text{Softmax}(\frac{\mathbf{q} \cdot \mathbf{k}^T}{\sqrt{d}}), \mathbf{m}_V = \text{Softmax}(-\frac{\mathbf{q} \cdot \mathbf{k}^T}{\sqrt{d}}) \tag{6}$$

where $d$ denotes feature dimension, $\mathbf{m}_I$ and $\mathbf{m}_V$ represent the masks of invariant and variant structural patterns. In this way, dynamic neighbors with higher attention scores in invariant patterns will have lower attention scores in variant ones, which means the invariant and variant patterns have a negative correlation. To capture invariant featural pattern, we adopt a learnable featural mask $\mathbf{m}_f = \text{Softmax}(\mathbf{w}_f)$ to select features from the messages of dynamic neighbors. Then the messages of dynamic neighborhood can be summarized with respective masks,

$$\begin{aligned} \mathbf{z}_I^t(u) &= \text{Agg}_I(\mathbf{m}_I, \mathbf{v} \odot \mathbf{m}_f) \\ \mathbf{z}_V^t(u) &= \text{Agg}_V(\mathbf{m}_V, \mathbf{v}) \end{aligned} \tag{7}$$

where $\text{Agg}(\cdot)$ denotes aggregating and summarizing messages from dynamic neighborhood. To further disentangle the invariant and variant patterns, we design different aggregation functions $\text{Agg}_I(\cdot)$ and $\text{Agg}_V(\cdot)$ to summarize specific messages from masked dynamic neighborhood respectively. Then the pattern summarizations are added up as hidden embeddings to be fed into subsequent layers.

$$\mathbf{h}_u^t \leftarrow \mathbf{z}_I^t(u) + \mathbf{z}_V^t(u) \tag{8}$$

**Overall Architecture.** The overall architecture is a stacking of spatio-temporal graph attention layers. Like classic graph message-passing networks, this enables each node to access high-order dynamic neighborhood indirectly, where $\mathbf{z}_I^t(u)$ and $\mathbf{z}_V^t(u)$ at $l$-th layer can be a summarization of invariant and variant patterns in $l$-order dynamic neighborhood. In practice, the attention can be easily extended to multi-head attention [42] to stable the training process and model multi-faceted graph evolution [43].

## 3.3 Spatio-Temporal Intervention Mechanism

**Direct Intervention.** One way of intervening variant pattern distribution as Eq. (4) is directly generating and altering the variant patterns. However, this is infeasible in practice due to the following reasons: First, since it has to intervene the dynamic neighborhood and features node-wisely, the computational complexity is unbearable. Second, generating variant patterns including time-varying structures and features is another intractable problem.

**Approximate Intervention.** To tackle the problems mentioned above, we propose to approximate the patterns $\mathbf{P}^t$ with summarized patterns $\mathbf{z}^t$ found in Sec. 3.2. As $\mathbf{z}_I^t(u)$ and $\mathbf{z}_V^t(u)$ act as summarizations of invariant and variant spatio-temporal patterns for node $u$ at time $t$, we approximate the intervention process by sampling and replacing the variant pattern summarizations instead of altering original structures and features with generated ones. To do spatio-temporal intervention, we collect variant patterns of all nodes at all time, from which we sample one variant pattern to replace the variant patterns of other nodes across time. For example, we can use the variant pattern of node $v$ at time $t_2$ to replace the variant pattern of node $u$ at time $t_1$ as

$$\mathbf{z}_I^{t_1}(u), \mathbf{z}_V^{t_1}(u) \leftarrow \mathbf{z}_I^{t_1}(u), \mathbf{z}_V^{t_2}(v) \tag{9}$$

As the invariant pattern summarization is kept the same, the label should not be changed. Thanks to the disentangled spatio-temporal graph attention, we get variant patterns across neighborhoods and time, which can act as natural intervention samples inside data so that the complexity of the generation problem can also be avoided. By doing Eq. (9) multiple times, we can obtain multiple intervened data distributions for the subsequent optimization.

### 3.4 Optimization with Invariance Loss

Based on the multiple intervened data distributions with different variant patterns, we can next optimize the model to focus on invariant patterns to make predictions. Here, we introduce invariance loss to instantiate Eq. (4). Let $\mathbf{z}_I$ and $\mathbf{z}_V$ be the summarized invariant and variant patterns, we calculate the task loss by only using the invariant patterns

$$\mathcal{L} = \ell(f(\mathbf{z}_I), \mathbf{y}) \tag{10}$$

where $f(\cdot)$ is the predictor. The task loss let the model utilize the invariant patterns to make predictions. Then we calculate the mixed loss as

$$\mathcal{L}_m = \ell(g(\mathbf{z}_V, \mathbf{z}_I), \mathbf{y}) \tag{11}$$

where another predictor $g(\cdot)$ makes predictions using both invariant patterns $\mathbf{z}_V$ and variant patterns $\mathbf{z}_I$. The mixed loss measure the model's prediction ability when variant patterns are also exposed to the model. Then the invariance loss is calculated by

$$\mathcal{L}_{do} = \text{Var}_{s_i \in \mathcal{S}}(\mathcal{L}_m | \text{do}(\mathbf{P}_V^t = s_i)) \tag{12}$$

where 'do' denotes the intervention mechanism as mentioned in Section. 3.3. The invariance loss measures the variance of the model's prediction ability under multiple intervened distributions. The final training objective is

$$\min_\theta \mathcal{L} + \lambda \mathcal{L}_{do} \tag{13}$$

where the task loss $\mathcal{L}$ is minimized to exploit invariant patterns while the invariance loss $\mathcal{L}_{do}$ helps the model to discover invariant and variant patterns, and $\lambda$ is a hyperparameter to balance between two objectives. After training, we only adopt invariant patterns to make predictions in the inference stage. The overall algorithm is summarized in Table 1.

---

**Algorithm 1** Training pipeline for **DIDA**

---

**Require:** Training epochs $L$, number of intervention samples $S$, hyperparameter $\lambda$
 1: **for** $l = 1, \ldots, L$ **do**
 2:     Obtain $\mathbf{z}_V^t, \mathbf{z}_I^t$ for each node and time as described in Section 3.2
 3:     Calculate task loss and mixed loss as Eq. (10) and Eq. (11)
 4:     Sample $S$ variant patterns from collections of $\mathbf{z}_V^t$, to construct intervention set $\mathcal{S}$
 5:     **for** $s$ in $\mathcal{S}$ **do**
 6:         Replace the nodes' variant pattern summarizations with $s$ as Section 3.3
 7:         Calculate mixed loss as Eq. (11)
 8:     **end for**
 9:     Calculate invariance loss as Eq. (12)
10:     Update the model according to Eq. (13)
11: **end for**

---

## 4 Experiments

In this section, we conduct extensive experiments to verify that our framework can handle spatio-temporal distribution shifts by discovering and utilizing invariant patterns. More Details of the settings and other results can be found in Appendix.

**Baselines.** We adopt several representative GNNs and Out-of-Distribution(OOD) generalization methods as our baselines:

- Static GNNs: **GAE** [44], a representative static GNN with stacking of graph convolutions; **VGAE** [44] further introduces variational variables into GAE.

- Dynamic GNNs: **GCRN** [45],a representative dynamic GNN that first adopts a GCN[44] to obtain node embeddings and then a GRU [46] to model the dynamics; **EvolveGCN** [13] adopts a LSTM[47] or GRU [46] to flexibly evolve the GCN[44] parameters instead of directly learning the temporal node embeddings; **DySAT** [43] models dynamic graph using structural and temporal self-attention.
- OOD generalization methods: **IRM** [48] aims at learning an invariant predictor which minimizes the empirical risks for all training domains; **GroupDRO** [49] reduces differences in risk across training domains to reduce the model's sensitivity to distributional shifts; **V-REx** [50] puts more weight on training domains with larger errors when minimizing empirical risk.

## 4.1 Real-world Datasets

**Settings.** We use 3 real-world dynamic graph datasets, including COLLAB, Yelp and Transaction. We adopt the challenging inductive future link prediction task, where the model exploits past graphs to make link prediction in the next time step. Each dataset can be split into several partial dynamic graphs based on its field information. For brevity, we use 'w/ DS' and 'w/o DS' to represent test data with and without distribution shift respectively. To measure models' performance under spatio-temporal distribution shift, we choose one field as 'w/ DS' and the left others are further split into training, validation and test data ('w/o DS') chronologically. Note that the 'w/o DS' is a merged dynamic graph without field information and 'w/ DS' is unseen during training, which is more practical and challenging in real-world scenarios. More details on their spatio-temporal distribution shifts are provided in Appendix. Here we briefly introduce the real-world datasets as follows

- **COLLAB** [51][4] is an academic collaboration dataset with papers that were published during 1990-2006. Node and edge represent author and coauthorship respectively. Based on the field of co-authored publication, each edge has the field information including "Data Mining", "Database", "Medical Informatics", "Theory" and "Visualization". The time granularity is year, including 16 time slices in total. We use "Data Mining" as 'w/ DS' and the left as 'w/o DS'.
- **Yelp** [43][5] is a business review dataset, containing customer reviews on business. Node and edge represent customer/business and review behavior respectively. We consider interactions in five categories of business including "Pizza", "American (New) Food", "Coffee & Tea ", "Sushi Bars" and "Fast Food" from January 2019 to December 2020. The time granularity is month, including 24 time slices in total. We use "Pizza" as 'w/ DS' and the left as 'w/o DS'.
- **Transaction**[6] is a secondary market transaction dataset, which records transaction behaviors of users from 10th April 2022 to 10th May 2022. Node and edge represent user and transaction respectively. The transactions have 4 categories, including "Pants", "Outwears", "Shirts" and "Hoodies". The time granularity is day, including 30 time slices in total. We use "Pants" as 'w/ DS' and the left as 'w/o DS'.

**Results.** Based on the results on real-world datasets in Table. 1, we have the following observations:

- Baselines fail dramatically under distribution shift: 1) Although DyGNN baselines perform well on test data without distribution shift, their performance drops greatly under distribution shift. In particular, the performance of DySAT, which is the best-performed DyGNN in 'w/o DS', drop by nearly 12%, 12% and 5% in 'w/ DS'. In Yelp and Transaction, GCRN and EGCN even underperform static GNNs, GAE and VGAE. This phenomenon shows that the existing DyGNNs may exploit variant patterns and thus fail to handle distribution shift. 2) Moreover, as generalization baselines are not specially designed to consider spatio-temporal distribution shift in dynamic graphs, they only have limited improvements in Yelp and Transaction. In particular, they rely on ground-truth environment labels to achieve OOD generalization, which are unavailable for real dynamic graphs. The inferior performance indicates that they cannot generalize well without accurate environment labels, which verifies that lacking environmental labels is also a key challenge for handling distribution shifts of dynamic graphs.
- Our method can better handle distribution shift than the baselines, especially in stronger distribution shift. **DIDA** improves significantly over all baselines in 'w/ DS' for all datasets. Note that

---

[4]https://www.aminer.cn/collaboration.

[5]https://www.yelp.com/dataset

[6]Collected from Alibaba.com

Table 1: Results(AUC%) of different methods on real-world datasets. The best results are in bold and the second-best results are underlined. 'w/o DS' and 'w/ DS' denote test data with and without distribution shift.

| Model | COLLAB | | Yelp | | Transaction | |
|---|---|---|---|---|---|---|
| Test Data | w/o DS | w/ DS | w/o DS | w/ DS | w/o DS | w/ DS |
| GAE | $77.15_{\pm0.50}$ | $74.04_{\pm0.75}$ | $70.67_{\pm1.11}$ | $64.45_{\pm5.02}$ | $71.90_{\pm0.32}$ | $73.44_{\pm0.41}$ |
| VGAE | $86.47_{\pm0.04}$ | $74.95_{\pm1.25}$ | $76.54_{\pm0.50}$ | $65.33_{\pm1.43}$ | $79.31_{\pm0.37}$ | $75.66_{\pm0.30}$ |
| GCRN | $82.78_{\pm0.54}$ | $69.72_{\pm0.45}$ | $68.59_{\pm1.05}$ | $54.68_{\pm7.59}$ | $78.99_{\pm0.28}$ | $71.24_{\pm0.35}$ |
| EGCN | $86.62_{\pm0.95}$ | $76.15_{\pm0.91}$ | $78.21_{\pm0.03}$ | $53.82_{\pm2.06}$ | $73.22_{\pm1.11}$ | $66.49_{\pm0.97}$ |
| DySAT | $\underline{88.77}_{\pm0.23}$ | $\underline{76.59}_{\pm0.20}$ | $78.87_{\pm0.57}$ | $66.09_{\pm1.42}$ | $81.55_{\pm0.66}$ | $76.18_{\pm0.43}$ |
| IRM | $87.96_{\pm0.90}$ | $75.42_{\pm0.87}$ | $66.49_{\pm10.78}$ | $56.02_{\pm16.08}$ | $81.65_{\pm0.50}$ | $75.61_{\pm0.61}$ |
| VREx | $88.31_{\pm0.32}$ | $76.24_{\pm0.77}$ | $\underline{79.04}_{\pm0.16}$ | $66.41_{\pm1.87}$ | $\underline{81.72}_{\pm0.35}$ | $\underline{76.24}_{\pm0.52}$ |
| GroupDRO | $88.76_{\pm0.12}$ | $76.33_{\pm0.29}$ | $\mathbf{79.38}_{\pm0.42}$ | $\underline{66.97}_{\pm0.61}$ | $81.50_{\pm0.24}$ | $75.92_{\pm0.37}$ |
| **DIDA** | $\mathbf{91.97}_{\pm0.05}$ | $\mathbf{81.87}_{\pm0.40}$ | $78.22_{\pm0.40}$ | $\mathbf{75.92}_{\pm0.90}$ | $\mathbf{83.08}_{\pm0.33}$ | $\mathbf{77.61}_{\pm0.59}$ |

Table 2: Results(AUC%) of different methods on synthetic dataset. The best results are in bold and the second-best results are underlined. Larger $\overline{p}$ denotes higher distribution shift level.

| Model $\backslash \overline{p}$ | 0.4 | | 0.6 | | 0.8 | |
|---|---|---|---|---|---|---|
| Split | Train | Test | Train | Test | Train | Test |
| GCRN | $69.60_{\pm1.14}$ | $\underline{72.57}_{\pm0.72}$ | $74.71_{\pm0.17}$ | $\underline{72.29}_{\pm0.47}$ | $75.69_{\pm0.07}$ | $\underline{67.26}_{\pm0.22}$ |
| EGCN | $78.82_{\pm1.40}$ | $69.00_{\pm0.53}$ | $79.47_{\pm1.68}$ | $62.70_{\pm1.14}$ | $81.07_{\pm4.10}$ | $60.13_{\pm0.89}$ |
| DySAT | $84.71_{\pm0.80}$ | $70.24_{\pm1.26}$ | $89.77_{\pm0.32}$ | $64.01_{\pm0.19}$ | $94.02_{\pm1.29}$ | $62.19_{\pm0.39}$ |
| IRM | $\underline{85.20}_{\pm0.07}$ | $69.40_{\pm0.09}$ | $89.48_{\pm0.22}$ | $63.97_{\pm0.37}$ | $\mathbf{95.02}_{\pm0.09}$ | $62.66_{\pm0.33}$ |
| VREx | $84.77_{\pm0.84}$ | $70.44_{\pm1.08}$ | $89.81_{\pm0.21}$ | $63.99_{\pm0.21}$ | $94.06_{\pm1.30}$ | $62.21_{\pm0.40}$ |
| GroupDRO | $84.78_{\pm0.85}$ | $70.30_{\pm1.23}$ | $\underline{89.90}_{\pm0.11}$ | $64.05_{\pm0.21}$ | $\underline{94.08}_{\pm1.33}$ | $62.13_{\pm0.35}$ |
| **DIDA** | $\mathbf{87.92}_{\pm0.92}$ | $\mathbf{85.20}_{\pm0.84}$ | $\mathbf{91.22}_{\pm0.59}$ | $\mathbf{82.89}_{\pm0.23}$ | $92.72_{\pm2.16}$ | $\mathbf{72.59}_{\pm3.31}$ |

Yelp has stronger temporal distribution shift since COVID-19 happens in the midway, strongly affecting consumers' behavior in business, while **DIDA** outperforms the most competitive baseline GroupDRO by 9% in 'w/ DS'. In comparison to similar field information in Yelp (all restaurants) and Transaction (all costumes), COLLAB has stronger spatial distribution shift since the fields are more different to each other, while **DIDA** outperforms the most competitive baseline DySAT by 5% in 'w/ DS'.

## 4.2 Synthetic Dataset

**Settings.** To evaluate the model's generalization ability under spatio-temporal distribution shift, following [18], we introduce manually designed shifts in dataset COLLAB with all fields merged. Denote original features and structures as $\mathbf{X}_1^t \in \mathbb{R}^{N \times d}$ and $\mathbf{A}^t \in \{0,1\}^{N \times N}$. For each time $t$, we uniformly sample $p(t)|\mathcal{E}^{t+1}|$ positive links and $(1-p(t))|\mathcal{E}^{t+1}|$ negative links in $\mathbf{A}^{t+1}$. Then they are factorized into variant features $\mathbf{X}_2^t \in \mathbb{R}^{N \times d}$ with property of structural preservation. Two portions of features are concatenated as $\mathbf{X}^t = [\mathbf{X}_1^t, \mathbf{X}_2^t]$ as input node features for training and inference. The sampling probability $p(t) = \mathrm{clip}(\overline{p} + \sigma cos(t), 0, 1)$ refers to the intensity of shifts, where the variant features $\mathbf{X}_2^t$ constructed with higher $p(t)$ will have stronger correlations with future link $\mathbf{A}^{t+1}$. We set $\overline{p}_{test} = 0.1, \sigma_{test} = 0, \sigma_{train} = 0.05$ and vary $\overline{p}_{train}$ in from 0.4 to 0.8 for evaluation. Since the correlations between $\mathbf{X}_2^t$ and label $\mathbf{A}^{t+1}$ vary through time and neighborhood, patterns include $\mathbf{X}_2^t$ are variant under distribution shifts. As static GNNs can not support time-varing features, we omit their results.

**Results.** Based on the results on synthetic dataset in Table. 2, we have the following observations:

- Our method can better handle distribution shift than the baselines. Although the baselines achieve high performance when training, their performance drop drastically in the test stage, which

shows that the existing DyGNNs fail to handle distribution shifts. In terms of test results, **DIDA** consistently outperforms DyGNN baselines by a significantly large margin. In particular, **DIDA** surpasses the best-performed baseline by nearly 13%/10%/5% in test results for different shift levels. For the general OOD baselines, they reduce the variance in some cases while their improvements are not significant. Instead, **DIDA** is specially designed for dynamic graphs and can exploit the invariant spatio-temporal patterns to handle distribution shift.

- Our method can exploit invariant patterns to consistently alleviate harmful effects of variant patterns under different distribution shift levels. As shift level increases, almost all baselines increase in train results and decline in test results. This phenomenon shows that as the relationship between variant patterns and labels goes stronger, the existing DyGNNs become more dependent on the variant patterns when training, causing their failure in test stage. Instead, the rise in train results and drop in test results of **DIDA** are significantly lower than baselines, which demonstrates that **DIDA** can exploit invariant patterns and alleviate the harmful effects of variant patterns under distribution shift.

### 4.3   Complexity Analysis

We analyze the computational complexity of **DIDA** as follows. Denote $|V|$ and $|E|$ as the total number of nodes and edges in the graph, respectively, and $d$ as the dimensionality of the hidden representation. The spatio-temporal aggregation has a time complexity of $O(|E|d + |V|d^2)$. The disentangled component adds a constant multiplier 2, which does not affect the time complexity of aggregation. Denote $|E_p|$ as the number of edges to predict and $|S|$ as the size of the intervention set. Our intervention mechanism has a time complexity of $O(|E_p||S|d)$ in training, and does not put extra time complexity in inference. Therefore, the overall time complexity of **DIDA** is $O(|E|d + |V|d^2 + |E_p||S|d)$. Notice that $|S|$ is a hyper-parameter and is usually set as a small constant. In summary, **DIDA** has a linear time complexity with respect to the number of nodes and edges, which is on par with the existing dynamic GNNs.

### 4.4   Ablation study

In this section, we conduct ablation studies to verify the effectiveness of the proposed spatio-temporal intervention mechanism and disentangled graph attention in **DIDA**.

**Spatio-temporal intervention mechanism.**
We remove the intervention mechanism mentioned in Sec 3.3. From Figure 2, we can see that without spatio-temporal intervention, the model's performance drop significantly especially in the synthetic dataset, which verifies that our intervention mechanism helps the model to focus on invariant patterns to make predictions.

**Disentangled graph attention.** We further remove the disentangled attention mentioned in Sec 3.2. From Figure 2, we can see that disentangled attention is a critical component in the model design, especially in Yelp dataset. Moreover, without disentangled module, the model is unable to obtain variant and invariant patterns for the subsequent intervention.

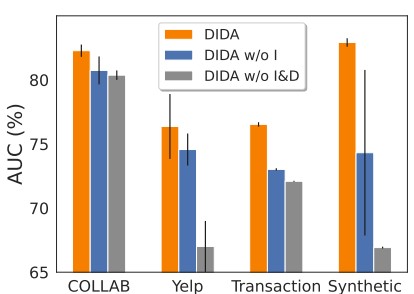

Figure 2: Ablation studies on intervention mechanism and disentangled attention, where 'w/o I' denotes removing the spatio-temporal intervention mechanism in **DIDA** and 'w/o I&D' further removes disentangled attention.

## 5   Related Work

**Dynamic Graph Neural Networks.** To tackle the complex structural and temporal information in dynamic graphs, considerable research attention has been devoted to dynamic graph neural networks (DyGNNs) [7, 8]. A classic of DyGNNs first adopt a GNN to aggregate structural information for graph at each time, followed by a sequence model like RNN [52, 53, 54, 45] or temporal self-attention [43] to process temporal information. Another classic of DyGNNs first introduce

time-encoding techniques to represent each temporal link as a function of time, followed by a spatial module like GNN or memory module [20, 55, 40, 41] to process structural information. To obtain more fine-grained continous node embeddings in dynamic graphs, some work further leverages neural interaction processes [56] and ordinary differential equation [57]. DyGNNs have been widely applied in real-world applications, including dynamic anomaly detection [58], event forecasting [59], dynamic recommendation [60], social character prediction [61], user modeling [62], temporal knowledge graph completion [63], etc. In this paper, we consider DyGNNs under spatio-temporal distribution shift, which remains unexplored in dynamic graph neural networks literature.

**Out-of-Distribution Generalization.** Most existing machine learning methods assume that the testing and training data are independent and identically distributed, which is not guaranteed to hold in many real-world scenarios [64]. In particular, there might be uncontrollable distribution shifts between training and testing data distribution, which may lead to sharp drop of model performance. To solve this problem, Out-of-Distribution (OOD) generalization problem has recently become a central research topic in various areas [65, 64, 66]. Recently, several works attempt to handle distribution shift on graphs [67, 29, 18, 68, 11, 69, 70, 71, 72, 73]. Another classic of OOD methods most related to our works handle distribution shifts on time-series data [25, 26, 12, 27, 28, 74]. Current works consider either only structural distribution shift for static graphs or only temporal distribution shift for time-series data. However, spatio-temporal distribution shifts in dynamic graphs are more complex yet remain unexplored. To the best of our knowledge, this is the first study of spatio-temporal distribution shifts in dynamic graphs.

**Disentangled Representation Learning.** Disentangled representation learning aims to characterize the multiple latent explanatory factors behind the observed data, where the factors are represented by different vectors [75]. Besides its applications in computer vision [76, 77, 78, 79, 80] and recommendation [81, 82, 83, 84, 85, 86], several disentangled GNNs have proposed to generalize disentangled representation learning in graph data recently. DisenGCN [87] and IPGDN [88] utilize the dynamic routing mechanism to disentangle latent factors for node representations. FactorGCN [89] decomposes the input graph into several interpretable factor graphs. DGCL [90, 91] aim to learn disentangled graph-level representations with self-supervision. Some works factorize deep generative models based on node, edge, static, dynamic factors [92] or spatial, temporal, graph factors [93] to achieve interpretable dynamic graph generation.

## 6    Conclusion

In this paper, we propose Disentangled Intervention-based Dynamic Graph Attention Networks (**DIDA**) to handle spatio-temporal distribution shift in dynamic graphs. First, we propose a disentangled dynamic graph attention network to capture invariant and variant spatio-temporal patterns. Then, based on the causal inference literature, we design a spatio-temporal intervention mechanism to create multiple intervened distributions and propose an invariance regularization term to help the model focus on invariant patterns under distribution shifts. Extensive experiments on three real-world datasets and one synthetic dataset demonstrate that our method can better handle spatio-temporal distribution shift than state-of-the-art baselines. One limitation is that in this paper we mainly consider dynamic graphs in scenarios of discrete snapshots, and we leave studying spatio-temporal distribution shifts in continous dynamic graphs for further explorations.

## Acknowledgements

This work was supported in part by the National Key Research and Development Program of China No. 2020AAA0106300, National Natural Science Foundation of China (No. 62250008, 62222209, 62102222, 62206149), China National Postdoctoral Program for Innovative Talents No. BX20220185 and China Postdoctoral Science Foundation No. 2022M711813. All opinions, findings, conclusions and recommendations in this paper are those of the authors and do not necessarily reflect the views of the funding agencies.

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
