# Dynamic Graph Neural Networks Under Spatio-Temporal Distribution Shift
## (Appendix)

## A  Notations

| Notations | Descriptions |
|---|---|
| $\mathcal{G} = (\mathcal{V}, \mathcal{E})$ | A graph with the node set and edge set |
| $\mathcal{G}^t = (\mathcal{V}^t, \mathcal{E}^t)$ | Graph slice at time $t$ |
| $\mathbf{X}^t, \mathbf{A}^t$ | Features and adjacency matrix of a graph at time $t$ |
| $\mathcal{G}^{1:t}, Y^t, \mathbf{G}^{1:t}, \mathbf{Y}^t$ | Graph trajectory, label and their corresponding random variable |
| $\mathcal{G}_v^{1:t}, y^t, \mathbf{G}_v^{1:t}, \mathbf{y}^t$ | Ego-graph trajectory, the node's label and their corresponding random variable |
| $f(\cdot), g(\cdot)$ | Predictors |
| $P, \mathbf{P}$ | Pattern and its corresponding random variable |
| $m(\cdot)$ | Function to select structures and features from ego-graph trajectories |
| $\mathrm{do}(\cdot)$ | do-calculus |
| $\phi(\cdot)$ | Function to find invariant patterns |
| d | The dimensionality of node representation |
| $\mathbf{q}, \mathbf{k}, \mathbf{v}$ | Query, key and value vector |
| $\mathcal{N}^t(u)$ | Dynamic neighborhood of node $u$ at time $t$ |
| $\mathbf{m}_I, \mathbf{m}_V, \mathbf{m}_f$ | Structural mask of invariant and variant patterns, and featural mask |
| $\mathbf{z}_I^t(u), \mathbf{z}_V^t(u)$ | Summarizations of invariant and variant patterns for node $u$ at time $t$ |
| $\mathrm{Agg}_I(\cdot), \mathrm{Agg}_V(\cdot)$ | Aggregation functions for invariant and variant patterns |
| $\mathbf{h}_u^t$ | Hidden embeddings for node $u$ at time $t$ |
| $\ell$ | Loss function |
| $\mathcal{L}, \mathcal{L}_m, \mathcal{L}_{do}$ | Task loss, mixed loss and invariance loss |

## B  More Details on Section 3.1

**Background of Assumption 1.** It is widely adopted in out-of-distribution generalization literature [1, 2, 3, 4, 5, 6, 7] about the assumption that the relationship between labels and some parts of features is invariant across data distributions, and these subsets of features with such properties are called invariant features. In this paper, we use invariant patterns $\mathbf{P}_I$ to denote the invariant structures and features. From the causal perspective, we can formulate the data-generating process in dynamic graphs with a structural causal model (SCM) [8, 9], $\mathbf{P}_V \rightarrow \mathbf{G} \leftarrow \mathbf{P}_I \rightarrow \mathbf{y}$ and $\mathbf{P}_V \leftarrow \mathbf{P}_I$, where the arrow between variables denotes casual relationship, and the subscript $v$ and superscript $t$ are omitted for brevity. $\mathbf{P}_V \rightarrow \mathbf{G} \leftarrow \mathbf{P}_I$ denotes that variant and invariant patterns construct the ego-graph trajectories observed in the data, while $\mathbf{P}_I \rightarrow \mathbf{y}$ denotes that invariant patterns determine the ground truth label $\mathbf{y}$, no matter how the variant patterns change inside data across different distributions. Sometimes, the correlations between variant patterns and labels may be built by some exogenous factors like periods and communities. In some distributions, $\mathbf{P}_V \leftarrow \mathbf{P}_I$ would open a backdoor path [9] $\mathbf{P}_V \leftarrow \mathbf{P}_I \rightarrow \mathbf{y}$ so that variant patterns $\mathbf{P}_V$ and labels $\mathbf{y}$ are correlated statistically, and this correlation is also called spurious correlation. If the model highly relies on the relationship between variant patterns and labels, it will fail under distribution shift, since such relationship varies across

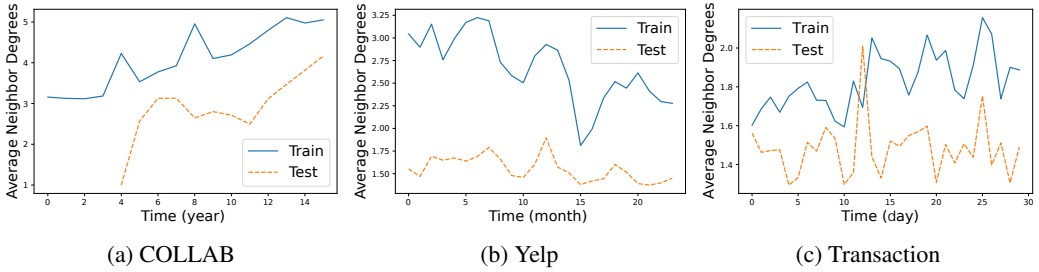

Figure 1: Average neighbor degrees in the graph slice as time goes.

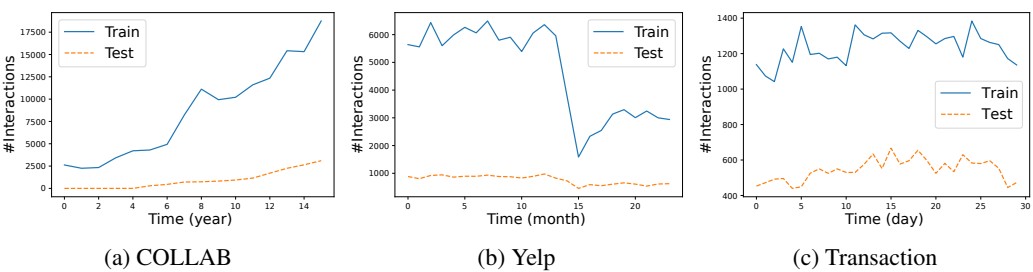

Figure 2: Number of links in the graph slice as time goes.

distributions. Hence, we propose to help the model focus on invariant patterns to make predictions and thus handle distribution shift.

**Connnections in Remark 1.** To eliminate the spurious correlation between variant patterns and labels, one way is to block the backdoor path by using do-calculus to intervene variant patterns. By applying do-calculus on one variable, all in-coming arrows(causal relationship) to it will be removed [9] and the intervened distributions will be created. In our case, the operator $do(\mathbf{P}_V)$ will cut the causal relationship from invariant patterns to variant patterns, i.e. disabling $\mathbf{P}_V \leftarrow \mathbf{P}_I$ and then blocking the backdoor path $\mathbf{P}_V \leftarrow \mathbf{P}_I \rightarrow \mathbf{y}$. Hence, the model can learn the direct causal effects from invariant patterns to labels in the intervened distributions $p(\mathbf{y}, \mathbf{G}|do(\mathbf{P}_V))$, and the risks should be the same across these intervened distributions. Therefore we can minimize the variance of empirical risks under different intervened distributions to help the model focus on the relationship between invariant patterns and labels. On the other hand, if we have the optimal predictor $f_{\theta_1}^*$ and pattern finder $\phi_{\theta_2}^*$ according to Eq.(3), then the variance term in Eq.(4) is minimized as the variant patterns will not affect the predictions of $f_{\theta_1}^* \circ \phi_{\theta_2}^*$ across different intervened distributions.

## C   Additional Experiments

### C.1   Distribution Shifts in Real-world Datasets

We illustrate the distribution shifts in the real-world datasets with two statistics, number of links and average neighbor degrees [10]. Figure 1 shows that the average neighbor degrees are lower in test data compared to training data. Lower average neighbor degree indicates that the nodes have less affinity to connect with high-degree neighbors. Moreover, in COLLAB, the test data has less history than training data, i.e. the graph trajectory is not always complete in training and test data distribution. This phenomenon of incomplete history is common in real-world scenarios, e.g. not all the users join the social platforms at the same time. Figure 2 shows that the number of links and its trend also differ in training and test data. In COLLAB, #links of test data has a slower rising trend than training data. In Yelp, #links of training and test data both have a drop during time 13-15 and rise again thereafter, due to the outbreak of COVID-19, which strongly affected the consumers' behavior.

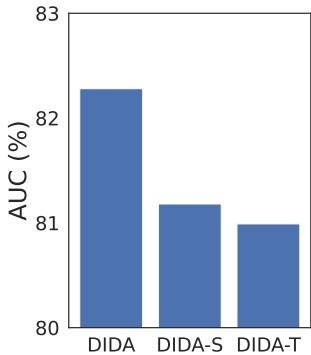

Figure 3: Comparison of different intervention mechansim on COLLAB dataset, where **DIDA-S** only uses spatial intervention and **DIDA-T** only uses temporal intervention.

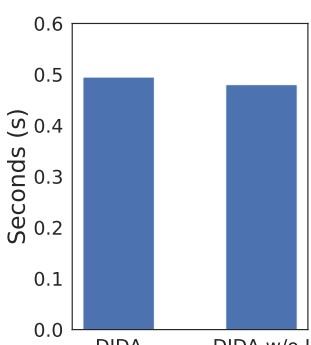

Figure 4: Comparison in terms of training time for each epoch on COLLAB dataset, where 'w/o I' means removing intervention mechanism in **DIDA**.

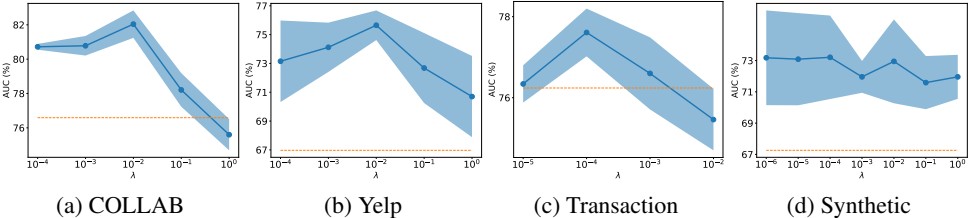

| (a) COLLAB | (b) Yelp | (c) Transaction | (d) Synthetic |

Figure 5: Sensitivity of hyperparameter $\lambda$. The area shows the average AUC and standard deviations in the test stage. The dashed line represents the average AUC of the best performed baseline.

## C.2 Spatial or Temporal Intervention

We compare two other versions of **DIDA** , where **DIDA-S** only uses spatial intervention and **DIDA-T** only uses temporal intervention. For **DIDA-S**, we put the constraint that the variant patterns used to intervene must come from the same timestamp in Eq.(9) so that the variant patterns across time are forbidden for intervention. Similarly, we put the constraint that the variant patterns used to intervene must come from the same node in Eq.(9) for **DIDA-T**. Figure 3 shows that **DIDA** improves significantly over the other two ablated versions, which verifies that it is important to take into consideration both the spatial and temporal aspects of distribution shifts.

## C.3 Efficiency of Intervention

For **DIDA** and **DIDA** without intervention mechanism, we compare their training time for each epoch on COLLAB dataset. As shown in Figure 4, the intervention mechanism adds few costs in training time (lower than 5%). Moreover, as **DIDA** does not use the intervention mechanism in the test stage, it does not add extra computational costs in the inference time.

## C.4 Hyperparameter Sensitivity

We analyze the sensitivity of hyperparameter $\lambda$ in **DIDA** for each dataset. From Figure 5, we can see that as $\lambda$ is too small or too large, the model's performance drops in most datasets. It shows that $\lambda$ acts as a balance between how **DIDA** exploits the patterns and satisfies the invariance constraint.

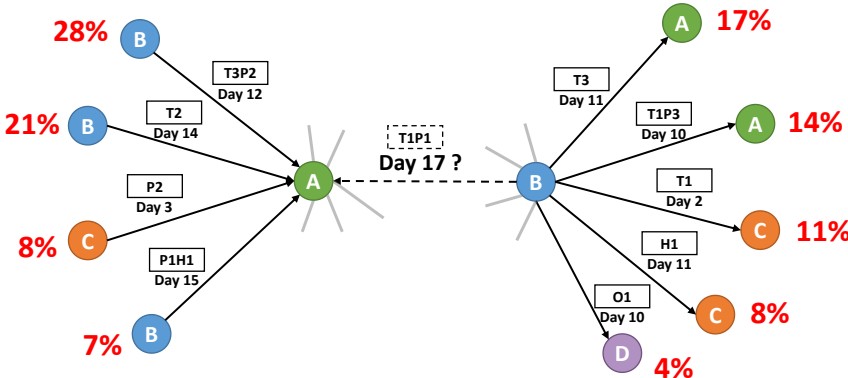

Figure 6: Case study: **DIDA**'s attention scores for invariant patterns, which are shown in percentage and marked red. Nodes and links represent users and transactions respectively. Links with smaller attention scores are omitted for brevity. For each link, the direction denotes a selling behavior, and it is tagged with the trading goods and transaction time on the link. For example, the link from B to A with tagging 'Day 12' and 'T3P2' represents that user B sells user A three T-shirts and two pants on day 12. The trading goods have four types: T, P, H, O represent T-shirt, Pants, Hoddie, and Outerwear respectively. Based on the transactions in their dynamic neighborhood, **DIDA** predicts whether user A will buy something from user B on day 17, and the dashed bounding box 'T1P1' refers to ground-truth trading goods.

## C.5 Case Study

Figure 6 illustrates **DIDA**'s attention scores for invariant patterns. We use a well-trained **DIDA** on Transaction dataset and show the scores of the structural mask for invariant patterns, i.e. $\mathbf{m}_I$ in Eq.(6). In this case, **DIDA** predicts whether user A will buy something ('T1P1') from user B on day 17, based on the transactions in their dynamic neighborhood. We have the following observations:

- Transactions with higher attention scores are more correlated with the transaction to predict. The transactions with attention scores 28% and 21% include goods 'T3P2' and 'T2', which are closed to the ground-truth trading goods 'T1P1' from B to A on day 17. On the right side, similarly, the transactions with attention scores 17%, 14% and 11% include goods 'T3', 'T1P3' and 'T1', which are closed to 'T1P1' as well. In contrast, the transactions with other unrelated goods like 'H1' and 'O1' have even smaller attention scores.

- Dynamic information is critical in the attention scores for invariant patterns. For the transactions with attention scores 21% and 8% on the left side, they include goods 'T2' and 'P2' which are both similar to 'T1P1'. However, the latter's attention score is much lower than the former's. This is because the latter happens much earlier than the former, and **DIDA** learns to attend to more recent transactions to capture users' recent interest.

These observations indicate that **DIDA** can summarize invariant patterns in dynamic neighborhood to capture the users' interests in trading (user A as a recent T-shirt/Pants buyer and user B as a recent T-shirt/Pants seller) and make predictions by matching the summarized interests.

## D  Reproducibility Details

### D.1  Training & Evaluation

**Hyperparameters.** For all methods, the hidden dimension is set to 16, the number of layers is set to 2. We adopt the Adam optimizer [11] with a learning rate 0.01, weight decay 5e-7 and set the patience of early stopping on the validation set as 50. For other hyperparameters, we adopt grid search for the best parameters using the validation split. For **DIDA**, we set the number of intervention samples as 1000 for all datasets, and $\lambda$ as 1e-2,1e-2,1e-4,1e-1 for COLLAB, Yelp, Transaction, and Synthetic dataset respectively.

Table 1: Summarization of dataset statistics.

| Dataset | # Timestamps | # Nodes | # Links | Temporal Granularity | Feature Dimension |
|---|---|---|---|---|---|
| COLLAB | 16 | 23035 | 151790 | year | 32 |
| Yelp | 24 | 13095 | 65375 | month | 32 |
| Transaction | 30 | 29526 | 53448 | day | 15 |

**Evaluation.** We randomly sample negative samples from nodes that do not have links, and the negative samples for validation and testing set are kept the same for all comparing methods. The number of negative samples is the same as positive ones. We use Area under the ROC Curve (AUC) as the evaluation metric. We use the inner product of the two learned node representations to predict links and use cross-entropy as the loss function $\ell$. We randomly run the experiments three times, and report the average results and standard deviations.

## D.2 Dataset Details

We summarize dataset statistics in Table 1 and describe dataset details as follows.

**COLLAB.** [12][1] We use word2vec [13] to extract 32-dimensional feature from paper abstracts and average to obtain author features. We use 10,1,5 chronological graph slices for training, validation and test respectively. The dataset includes 23035 nodes and 151790 links in total.

**Yelp.** [14][2] We use word2vec [13] to extract 32-dimensional feature from reviews and average to obtain user and business features. We select users and items with interactions of more than 10. We use 15,1,8 chronological graph slices for training, validation and test respectively. The dataset includes 13095 nodes and 65375 links in total.

**Transaction.**[3] We calculate the distribution of users' historical transaction categories as the initial 15-dimensional features. We use 20,2,8 chronological graph slices for training, validation and test respectively. The dataset includes 29526 nodes and 53448 links in total.

**Synthetic.** We use the same features as $\mathbf{X}_1^t$ and structures as $\mathbf{A}^t$ in COLLAB, and introduce features $\mathbf{X}_2^t$ with variable correlation with supervision signals. $\mathbf{X}_2^t$ are obtained by training the embeddings $\mathbf{X}_2 \in \mathbb{R}^{N \times d}$ with reconstruction loss $\ell(\mathbf{X}_2 \mathbf{X}_2^T, \tilde{\mathbf{A}}^{t+1})$, where $\tilde{\mathbf{A}}^{t+1}$ refers to the sampled links, and $\ell$ refers to cross-entropy loss function. The embeddings $\mathbf{X}_2^t$ are trained with Adam optimizer, learning rate 1e-1, weight decay 1e-5 and earlystop patience 50. In this way, we empirically find that the inner product predictor can achieve results of over 99% AUC by using $\mathbf{X}_2^t$ to predict the sampled links $\tilde{\mathbf{A}}^{t+1}$, so that the generated features can have strong correlations with the sampled links. By controlling the $p$ mentioned in the Section 4.2, we can control the correlations of $\mathbf{X}^t$ and labels $\mathbf{A}^{t+1}$ to vary in training and test stage.

## D.3 Baseline Details

**Backbones.** This class of methods aim at improving the modeling ability for dynamic graphs.

- **GAE.** [15] A representative static GNN with stacking of graph convolutions.
- **VGAE.** [15] A representative static GNN which introduces variational variables into GAE.
- **GCRN.** [16] A representative dynamic GNN that first adopts a GCN to obtain node embeddings and then a GRU to model the dynamics.
- **DySAT.** [14] A representative dynamic GNN that models dynamic graph using structural and temporal self-attention.
- **EvolveGCN.** [17] A representative dynamic GNN that uses an RNN to evolve the GCN parameters instead of directly learning the temporal node embeddings.

**OOD Generalization methods.** This class of methods aim at improving the robustness and generalization ability of models against distribution shift. For fair comparison, we randomly split the

---

[1]https://www.aminer.cn/collaboration.

[2]https://www.yelp.com/dataset

[3]Collected from Alibaba.com

samples into different domains, as the field information is unknown to all methods. Since they are general OOD generalization methods and are not specifically designed for dynamic graphs, we adopt DySAT as their backbone, which is the best-performed DyGNN on training dataset.

- **IRM** [2] aims at learning an invariant predictor which minimizes the empirical risks for all training domains.
- **VREx** [18] reduces differences in risk across training domains to reduce the model's sensitivity to distributional shifts.
- **GroupDRO** [19] puts more weight on training domains with larger errors when minimizing empirical risk.

### D.4 Details of DIDA

**First Layer.** Before stacking of disentangled spatio-temporal graph attention Layers, we use a fully-connected layer $FC(\cdot)$ to transform the features into hidden embeddings.

$$FC(\mathbf{x}) = \mathbf{W}\mathbf{x} + \mathbf{b} \tag{1}$$

**Agg Functions.** We implement the aggregation function for invariant and variant patterns as

$$\tilde{\mathbf{z}}_I^t(u) = \sum_i \mathbf{m}_{I,i}(\mathbf{v}_i \odot \mathbf{m}_f), \quad \mathbf{z}_I^t(u) = FFN(\tilde{\mathbf{z}}_I^t(u) + \mathbf{h}_u^t)$$

$$\tilde{\mathbf{z}}_V^t(u) = \sum_i \mathbf{m}_{V,i}\mathbf{v}_i, \quad \mathbf{z}_V^t(u) = FFN(\tilde{\mathbf{z}}_V^t(u)) \tag{2}$$

**FFN.** The FFN includes a layer normalization [20], multi-layer perceptron and skip connection.

$$FFN(\mathbf{x}) = \alpha \cdot MLP(LayerNorm(\mathbf{x})) + (1 - \alpha) \cdot \mathbf{x} \tag{3}$$

where $\alpha$ is a learnable parameter.

**Predictor.** For link prediction task, we implement the predictor $f(\cdot)$ in Eq.(10) as inner product of hidden embeddings, i.e. $f(\mathbf{z}_I^t(u), \mathbf{z}_I^t(v)) = \mathbf{z}_I^t(u) \cdot (\mathbf{z}_I^t(v))^T$, which is conformed to classic link prediction settings. To implement the predictor $g(\cdot)$ in Eq.(11), we adopt the biased training technique following [21], i.e. $g(\mathbf{z}_V^t(u), \mathbf{z}_I^t(u), \mathbf{z}_V^t(v), \mathbf{z}_I^t(v)) = f(\mathbf{z}_I^t(u), \mathbf{z}_I^t(v)) \cdot \sigma(f(\mathbf{z}_V^t(u), \mathbf{z}_V^t(v)))$.

### D.5 Configurations

Experiments on COLLAB, Yelp and Synthetic datasets are conducted with:

- Operating System: Ubuntu 18.04.1 LTS
- CPU: Intel(R) Xeon(R) Gold 6240R CPU @ 2.40GHz
- GPU: NVIDIA GeForce RTX 3090 with 24 GB of memory
- Software: Python 3.8.13, Cuda 11.3, PyTorch [22] 1.11.0, PyTorch Geometric [23] 2.0.3.

Experiments on Transaction dataset are conducted with:

- Operating System: Ubuntu 18.04.5 LTS
- CPU: Intel(R) Xeon(R) Platinum 8163 CPU @ 2.50GHz
- GPU: NVIDIA Tesla V100 with 16 GB of memory
- Software: Python 3.6.12, Cuda 10.1, PyTorch [22] 1.8.2, PyTorch Geometric [23] 2.0.3.