# OpenReview forum: "Dynamic Graph Neural Networks Under Spatio-Temporal Distribution Shift"
_NeurIPS.cc/2022/Conference — NeurIPS 2022 Accept_

### Official Review · Reviewer_nS4E · 2022-07-10

**Rating:** 7
**Confidence:** 4
**Soundness:** 3 good
**Presentation:** 3 good
**Contribution:** 3 good

**Summary:**

This paper studies the problem of spatio-temporal distribution shift in dynamic graphs.
By disentangling the patterns in dynamic graphs into invariant and variant ones, the invariant patterns are utilized for stable prediction and the impact of distribution shift can be reduced.
Although the distribution shift has been widely studied in the literature on computer vision and natural language processing, the authors made early attempts in the dynamic graph.
I am overall positive about this work.

**Questions:**

1. Can the proposed model generalized to the continuous dynamic graph?
2. What is the variant and invariant pattern in dynamic graph? Is there any common understanding rather than the specific graph type?

**Limitations:**

The core definition of variant and invariant are not well explained, which limits the generalization and scalability of the proposed method.


**Strengths And Weaknesses:**

Strengths

[+] The distribution shift in the dynamic graph is critical and the authors made the early attempts on this topic, which should be encouraged in the community.

[+] The proposed solution is technical sound, the disentanglement as well as the causal spatio-temporal intervention mechanism can satisfy the requirements.

[+] The experiments are extensive and the results are encouraging.

Weaknesses

[-] The invariant pattern is assumed to be dependent on the time. From my opinion of view, the invariant pattern can be futher divided into the time dependent and time independent ones.

[-] The distribution shift in the experimental datasets are manually conducted，it would be better to have some automatically designed mechanism.

---

> ### Author Response · Authors · 2022-08-02
> **Response to Reviewer nS4E**
>
> **Q1: The invariant pattern can be further divided into the time dependent and time independent ones.**
>
> A1: Thank you for your comment. We agree that further analyzing and dividing the invariant patterns as you suggest is an interesting idea. As our paper is the first work on studying spatio-temporal distribution shifts in dynamic GNNs, we leave such further explorations as promising future works.
>
> **Q2: The distribution shift in the experimental datasets are manually conducted, it would be better to have some automatically designed mechanism.**
>
> A2: Thank you for your suggestions. We agree that automatically designed mechanism to add distribution shifts are important for research on the generalization of dynamic GNNs. However, since there is currently no prior work on designing the generation mechanism of distribution shift on dynamic graphs, we propose this manual distribution shift in experiments, which we believe is simple yet reasonable. The experimental results also suggest that such manual distribution shifts can differentiate the generalization ability of different models to a certain extent. We leave designing more automated generation mechanisms for distribution shifts on dynamic graphs as future works.
>
> **Q3: Can the proposed model be generalized to the continuous dynamic graph?**
>
> A3: Thank you for your question. We agree that continuous dynamic graph is also an important research problem. As the first work to study spatio-temporal distribution shifts in dynamic GNNs, we currently focus on conducting experiments in discrete dynamic graphs. One possible extension of our method to continuous dynamic graphs may be adopting a continuous time-encoding technique and a continuous dynamic graph predictor, which we leave as future explorations.
>
> **Q4:What is the variant and invariant pattern in dynamic graph? Is there any common understanding rather than the specific graph type?**
>
> A4: Thank you for your question. Invariant patterns generally refer to parts of the data that are sufficiently predictive, whose relationships with labels are stable across distribution shifts. For dynamic graphs, we define invariant patterns as subsets of ego-graphs across time stamps whose predictivity to labels are stable across time periods and graph communities. Here we also provide some conceptual examples. In road networks, for example, two traffic jams in different places and times may happen simultaneously by chance or there can be causal relations, e.g., the road structure let one traffic jam to block other roads and inevitably lead to another traffic jam. Only the latter case forms invariant patterns and can be used for stable predictions. Take recommendation systems for another example. Users' purchase of a sequence of items may be correlational or there can exist stable and invariant patterns, e.g., first buy a main product and then buy the accessories of the main product. In the case study shown in Appendix C.5, we show that DIDA can summarize invariant patterns in the temporal and neighborhood structure to capture the users' interests in shopping and make predictions of future interactions by matching the summarized recent interests, leading to better generalization abilities.

---

### Official Review · Reviewer_8AMq · 2022-07-11

**Rating:** 5
**Confidence:** 2
**Soundness:** 2 fair
**Presentation:** 2 fair
**Contribution:** 2 fair

**Summary:**

This paper introduces a method of dynamic graph neural networks with spatio and temporal intervention mechanism.

**Questions:**

in equation (6), why are the expressions for m_i and m_v identital?


**Limitations:**

I failed to find the accurate definition of 'ego-graph', 'distribution shifts', 'invariant and variance structural patterns' , etc. As a result, it is not easy to understand this paper correctly without reading several previous papers.

**Strengths And Weaknesses:**

Strength:
(1) the empirical study shows considerable improvement on existing method.
(2) Innovative using attention layers to capture spatio-temporal information.

Weakness:
(1) no strict proof or detailed illustration to show why spatio-temporal intervention works.
(2) computational complexity is not discussed in the main contents.

---

> ### Author Response · Authors · 2022-08-02
> **Response to Reviewer 8AMq**
>
> **Q1: Strict proof or detailed illustration to show why spatio-temporal intervention works.**
>
> A1: Thank you for your comment. We would like to clarify how our proposed spatio-temporal intervention works. In short, our proposed method utilizes the do-calculus from causal theory to cut the backdoor path from variant patterns to labels and help the model to focus on the invariant patterns to labels. Based on the invariance literature, our proposed method can alleviate the harm of variant patterns under spatio-temporal distribution shifts and improve the generalization ability. We provide these analyses and some background knowledge of causal theory in Appendix B. We also give a case study to illustrate that DIDA learns to exploit invariant patterns to make predictions in Appendix C.5. We agree that strict theoretical analyses could further enhance our paper. Considering that this is the first work in studying spatio-temporal distribution shifts in dynamic graphs, we leave such explorations as promising future works.
>
> **Q2: Computational complexity is not discussed in the main contents.**
>
> A2: Thank you for your comment. Following your suggestions, we analyze the computational complexity of our proposed method as follows. Denote $|V|$ and $|E|$ as the total number of nodes and edges in the graph, respectively, and $d$ as the dimensionality of the hidden representation. The spatio-temporal aggregation has a time complexity of $O(|E|d+|V|d^2)$. The disentangled component adds a constant multiplier $2$, which does not affect the time complexity of aggregation. Denote $|E_p|$ as the number of edges to predict and $|S|$ as the size of the intervention set. Our intervention mechanism has a time complexity of $O(|E_p||S|d)$ in training, and does not put extra time complexity in inference. Therefore, the overall time complexity of our method is $O(|E|d+|V|d^2 + |E_p||S|d)$. Notice that $|S|$ is a hyper-parameter and is usually set as a small constant. In summary, our proposed method has a linear time complexity with respect to the number of nodes and edges, which is on par with the existing dynamic GNNs. Empirically, we also find that our intervention mechanism does not put much extra computational costs as shown in Appendix C.3. We will add this discussion in the revised version.
>
> **Q3: In equation (6), why are the expressions for m_i and m_v identical?**
>
> A3: Thank you for your comment. In the main paper, Eq. (6) is
>
> $$\mathbf{m}_{I}=\operatorname{Softmax}\left(\frac{\mathbf{q} \cdot \mathbf{k}^{T}}{\sqrt{d}}\right) $$
>
> $$\mathbf{m}_{V}=\operatorname{Softmax}\left(-\frac{\mathbf{q} \cdot \mathbf{k}^{T}}{\sqrt{d}}\right)$$
> , where it should be noticed that $\mathbf{m}_V$ and $\mathbf{m}_I$ differ in a minus sign in the Softmax function. Our design objective is to let dynamic neighbors with higher attention scores be in the invariant patterns, and let those with lower attention scores be in variant ones. Therefore, the invariant and variant patterns have a negative correlation and capture complementary information.
>
> **Q4: Failing to find the accurate definition of 'ego-graph', 'distribution shifts', 'invariant and variant structural patterns' , etc. As a result, it is not easy to understand this paper correctly without reading several previous papers.**
>
> A4: Thank you for your comments. We clarify these concepts as follows. 'distribution shifts' describes that the training and testing data distributions are inconsistent, i.e. $p_{train}(\mathbf{X},\mathbf{Y})\neq p_{test}(\mathbf{X},\mathbf{Y})$, so that minimizing empirical risks in the training datasets may not lead to good results in the test datasets. 'invariant patterns' generally refer to sufficiently predictive parts of the data whose relationships with labels across distribution shifts are stable. For dynamic graphs, we define 'invariant structural patterns' as a subset of ego-graphs across time stamps whose predictive patterns to labels are stable across time periods and graph communities, while 'variant structural patterns' are the complement of invariant structural patterns so that their relationships are unstable, i.e., spurious correlations. An ego-graph is formally defined as $\mathcal{G}_v=(\mathbf{X}_v,\mathbf{A}_v)$ where $\mathbf{A}_v$ is the adjacency matrix including all edges in node $v$'s $L$-hop neighbors $\mathcal{N}_v$ (where $L$ is an arbitrary integer) and $\mathbf{X}_v$ includes the features of nodes in $\mathcal{N}_v$. We will clarify these expressions in the revised version.

---

### Official Review · Reviewer_zQjJ · 2022-07-12

**Rating:** 7
**Confidence:** 4
**Soundness:** 3 good
**Presentation:** 3 good
**Contribution:** 3 good

**Summary:**

This paper investigates graph neural networks on dynamic graphs, especially under spatio-temporal distribution shifts. The authors recognize that distribution shift is an important factor for dynamic graph embedding, which is not well-handled by the existing approaches. To address this, the authors propose a novel model named DIDA, to handle spatio-temporal distribution shifts in dynamic graphs by discovering and fully utilizing invariant spatio-temporal patterns. Experiments on four datasets demonstrate the effectiveness of the proposed model.

**Questions:**

Please see the weaknesses.

**Strengths And Weaknesses:**

Strengths:

1. The paper is well-written and easy to follow.
2. Dealing with the spatio-temporal information on dynamic graphs from the perspective of discovering and utilizing invariant patterns, I feel might be an effective direction.
3. The experiments are sufficient to demonstrate the performance of the proposed model.


Weaknesses:

1. The related studies discussed in Related Work are not quite sufficient. I suggest the authors cite and discuss more.
2. It is better to give the details for the baselines (e.g., the differences with the proposed model), as well as more details for the datasets (e.g., give a statistic table) in the main paper.

---

> ### Author Response · Authors · 2022-08-02
> **Response to Reviewer zQjJ**
>
> **Q1: More related works**
>
> A1: Thank you for your suggestion. We will add more related works in the revised version.
>
> **Q2: Better to give the details for the baselines (e.g., the differences with the proposed model), as well as more details for the datasets (e.g., give a statistic table) in the main paper.**
>
> A2: Thank you for your comments. We agree that incorporating these details can further improve our paper. However, due to the page limit, currently we are only able to include them in the appendix. We will reorganize and move them from the appendix into the main paper when the page limit permits.

---

### Official Review · Reviewer_vUM5 · 2022-07-14

**Rating:** 6
**Confidence:** 4
**Soundness:** 3 good
**Presentation:** 2 fair
**Contribution:** 3 good

**Summary:**

This work studies the patio-temporal distribution shift issue of dynamic graph neural networks. To pursue the robustness of DyGNNs, the authors proposed a specific invariant learning method and conducted experiments on both real-world and synthetic datasets.

**Questions:**

1. The experiment results need more explanations. For instance, why IRM and GroupDRO achieve inferior performance under the "w/ DS" setting. Why the compared methods show different trends on the real-world and synthetic datasets, e.g., GCRN performs quite well on the synthetic datasets.
2. As compared to [18], what is the advantage of the proposed method.


**Limitations:**

No.

**Strengths And Weaknesses:**

Strong points:
1. This paper revels the impact of distribution drift in DyGNNs, which forms a new research problem.
2. This paper presents a new method for training distributionally robust DyGNNs.
3. Extensive experiments validate the effectiveness of the proposed method.

Weak points:
1. The experiment results need more explanations. For instance, why IRM and GroupDRO achieve inferior performance under the "w/ DS" setting. Why the compared methods show different trends on the real-world and synthetic datasets, e.g., GCRN performs quite well on the synthetic datasets.
2. As compared to [18], what is the advantage of the proposed method.

---

> ### Author Response · Authors · 2022-08-02
> **Response to Reviewer vUM5**
>
> **Q1.1:Why IRM and GroupDRO achieve inferior performance under the "w/ DS" setting?**
>
> A1.1: Thank you for your question. IRM and GroupDRO rely on ground-truth environment labels to achieve OOD generalization. Since they are unavailable for real dynamic graphs, we follow the literature and use random environment labels for IRM and GroupDRO in our experiments. The inferior performance indicates that IRM and GroupDRO cannot generalize well without accurate environment labels, which verifies that lacking environmental labels is a key challenge for handling distribution shifts of dynamic graphs. We will add this discussion in the revised version.
>
> **Q1.2:Why the compared methods show different trends on the real-world and synthetic datasets, e.g., GCRN performs quite well on the synthetic datasets?**
>
> A1.2: Thank you for your question. Compared to real-world datasets, synthetic datasets have manually designed distribution shifts. A plausible reason for the inconsistent performance of GCRN is that the model manages to capture the manually designed distribution shift in synthetic graphs, but fails to tackle the more complex distribution shifts in real-world datasets. We will add this discussion in the revised version.
>
> **Q2:As compared to [18], what is the advantage of the proposed method?**
>
> A2: Thank you for your comment. EERM [18] proposes multiple context explorers that are adversarially trained to maximize the variance of risks from multiple virtual environments so that the model can extrapolate from a single observed environment. It shows a strong generalization ability in node-level predictions. However, EERM is designed for static graphs, and can not be directly applied to dynamic graphs where spatial-temporal distribution shifts exist. In comparison, our method is specially designed for dynamic graph and achieve strong performance of tackling spatio-temporal distribution shifts on dynamic graphs. We will add this discussion in the revised version.

---

### Meta-Review · Area_Chair_gzDo · 2022-08-21

**Recommendation:** Accept
**Confidence:** Certain

**Metareview:**

The paper addresses spatio-temporal distribution shifts in dynamic graphs by discovering and utilizing invariant patterns, i.e., structures and features whose predictive abilities are stable across distribution shifts. The paper is an early try to address distribution shifts in dynamic graphs, which is an interesting and important problem. The experiments show the effectiveness of the proposed methods on synthetic and real-world graphs. The authors are strongly encouraged to add more discussion on the experiments and baselines, related work, definitions of 'ego-graph', 'distribution shifts', 'invariant and variance structural patterns', computational complexity, and the other clarifications requested by the reviewers in the final version.

**Award:**

No

---

### Decision · Program_Chairs · 2022-09-14

Accept